# Integrated Analysis of Single-Cell and Bulk RNA Sequencing Data Reveals Memory-like NK Cell Subset Associated with *Mycobacterium tuberculosis* Latency

**DOI:** 10.3390/cells13040293

**Published:** 2024-02-06

**Authors:** Mojtaba Shekarkar Azgomi, Giusto Davide Badami, Marianna Lo Pizzo, Bartolo Tamburini, Costanza Dieli, Marco Pio La Manna, Francesco Dieli, Nadia Caccamo

**Affiliations:** 1Central Laboratory of Advanced Diagnosis and Biomedical Research (CLADIBIOR), Azienda Ospedaliera Universitaria Policlinico (AOUP) Paolo Giaccone, University of Palermo, 90127 Palermo, Italy; mojtaba.shekarkarazgomi@unipa.it (M.S.A.); giustodavide.badami@unipa.it (G.D.B.); marianna.lopizzo@unipa.it (M.L.P.); bartolo.tamburini@unipa.it (B.T.); costanza.dieli@unipa.it (C.D.); marcopio.lamanna@unipa.it (M.P.L.M.); nadia.caccamo@unipa.it (N.C.); 2Department of Biomedicine, Neurosciences and Advanced Diagnostic (BND), University of Palermo, 90127 Palermo, Italy; 3Department of Health Promotion, Mother and Childcare, Internal Medicine and Medical Specialties, University of Palermo, 90129 Palermo, Italy

**Keywords:** tuberculosis, *Mycobacterium tuberculosis*, latent *Mycobacterium tuberculosis* infection, single-cell RNA sequence, NK cells, NKG2C

## Abstract

Natural killer (NK) cells are innate-like lymphocytes that belong to the family of type-1 innate lymphoid cells and rapidly respond to virus-infected and tumor cells. In this study, we have combined scRNA-seq data and bulk RNA-seq data to define the phenotypic and molecular characteristics of peripheral blood NK cells. While the role of NK cells in immune surveillance against virus infections and tumors has been well established, their contribution to protective responses to other intracellular microorganisms, such as *Mycobacterium tuberculosis* (Mtb), is still poorly understood. In this study, we have combined scRNA-seq data and bulk RNA-seq data to illuminate the molecular characteristics of circulating NK cells in patients with active tuberculosis (TB) disease and subjects with latent Mtb infection (LTBI) and compared these characteristics with those of healthy donors (HDs) and patients with non-TB other pulmonary infectious diseases (ODs). We show here that the NK cell cluster was significantly increased in LTBI subjects, as compared to patients with active TB or other non-TB pulmonary diseases and HD, and this was mostly attributable to the expansion of an NK cell population expressing *KLRC2*, *CD52*, *CCL5* and H*LA-DRB1*, which most likely corresponds to memory-like NK2.1 cells. These data were validated by flow cytometry analysis in a small cohort of samples, showing that LTBI subjects have a significant expansion of NK cells characterized by the prevalence of memory-like CD52^+^ NKG2C^+^ NK cells. Altogether, our results provide some new information on the role of NK cells in protective immune responses to Mtb.

## 1. Introduction

NK cells are innate-like lymphocytes that belong to the family of type-1 innate lymphoid cells and rapidly respond to virus-infected and tumor cells. Unlike T and B lymphocytes, NK cells lack clonally distributed antigen-specific receptors, and their target cell recognition and functional activities rely on germline-encoded killer activating receptors (KARs) and killer inhibitory receptors (KIRs) [1]. According to the “missing self” hypothesis, NK cells kill cells lacking MHC class I expression, which are ligands of KIRs, while cells expressing MHC class I molecules are spared. Simultaneously, tumor transformation or virus infections upregulate the expression of stress-related molecules, which are ligands of KARs on NK cells. In addition, NK cell activities may be modulated by stimulatory cytokines, such as IL-2, IL-12, IL-15 and IL-18, or by KAR-binding soluble ligands or by IgG-opsonized target cells recognized by the FcγRIII (CD16) receptor expressed on NK cells (a phenomenon known as antibody-dependent cell-mediated cytotoxicity (ADCC)). Upon interaction with target cells, NK cells perform cytolytic activity and secrete a variety of pro-inflammatory cytokines such as IFN-γ and TNF-α.

While the role of NK cells in immune surveillance against virus infections and tumors has been well established, their contribution to protective responses to other intracellular microorganisms, such as Mtb is still poorly understood. In mice, vaccination with Bacille Calmette–Guerin (BCG) activates NK cells to produce IL-22 and IFN-γ and inhibit Mtb multiplication [2]. This protective effect is dependent on the NK cell killing of extracellular Mtb and on cooperation with monocytes [3], γδ T cells [4] or CD8 T cells [5]. In another study on mice, BCG vaccination induced an IFN-γ-producing memory-like NK cell subset, which provided protection against Mtb [6]. In humans, BCG vaccination did not modify the frequency of NK cells but promoted their production of IFN-γ and other pro-inflammatory cytokines [7]. In human TB disease, NK cells have been detected within granulomatous lesions [8] and in pleural fluid [9,10], suggesting their participation in immune responses against Mtb at the site of infection/disease. Some studies have reported reduced percentages of circulating total NK cells [11,12], or their subsets [12,13], in patients with active TB disease—as compared with LTBI subjects—and HD, which normalize after successful therapy, suggesting a role in the control of Mtb infection [14]. Conversely, another study did not report statistically significant differences in the frequencies of circulating NK cells between patients with active TB disease, LTBI subjects and HD [15]. In addition, there are also contrasting results on the percentages of the CD56^bright^ subset of NK cells in active TB patients, which were reported reduced in one study [12] but increased in another [15]. Single-cell sequencing technology offers an unprecedented opportunity to deepen our understanding of the transcriptomic, genomic, proteomic, epigenomic and metabolomic information of single cells. Very recently, single-cell RNA sequencing (scRNA-seq) analysis has yielded promising, yet preliminary, information on immune phenotypes in TB [16], generally confirming a marked decrease in several lymphocyte subsets, including NK cells, in active TB patients.

In this study, we have combined scRNA-seq and bulk RNA-seq data to define the phenotypic and molecular characteristics of peripheral blood NK cells. We report here that LTBI subjects have a significant expansion of NK cells characterized by the prevalence of memory-like CD52^+^ NKG2C^+^ NK cells. This study may provide some new information on the role of NK cells in protective immune responses to Mtb.

## 2. Materials and Methods

### 2.1. Sample Collection

Between 1 October 2023 and 30 January 2024, Policlinico Palermo University Hospital in Italy conducted a prospective enrolment of adult patients with active TB. The study also included subjects with LTBI during this timeframe and HD. The diagnosis of active TB was based on clinical symptoms, chest radiography and microscopy for acid-fast bacilli (AFB), sputum Mtb culture and response to anti-TB chemotherapy. On the other hand, individuals with LTBI were identified by testing positive using QuantiFERON-TB Gold Plus and were characterized by lacking clinical symptoms and radiologic signs of active TB. Notably, individuals with human immunodeficiency virus infection (HIV) or other immunosuppressive conditions were excluded from the study. This study included a total of 36 participants (active TB [*n* = 12], LTBI [*n* = 12] and HD [*n* = 12]). Each participant contributed 6 mL of blood, collected in EDTA tube, with samples obtained prior to the initiation of anti-TB or TB preventive treatment. The collected blood underwent immediate processing to isolate peripheral blood mononuclear cells (PBMCs). Subsequently, the separated cells were meticulously preserved at −80 °C, maintaining their integrity until the analytical phase.

### 2.2. Data Collection

All datasets used in this study were retrieved from the National Center for Biotechnology Information’s (NCBI) Gene Expression Omnibus (GEO) database (https://www.ncbi.nlm.nih.gov/geo/ (accessed on 21 October 2023), a public repository for gene expression data. A search of the GEO profiles related to TB and LTBI samples in GEO database using the terms “Tuberculosis” [mesh terms] OR active tuberculosis [all fields] AND “Homo sapiens” [porgn] led to identifying 12 distinct studies (GSE37250, GSE39939, GSE39940, GSE40553, GSE42825, GSE42826, GSE42827, GSE42830, GSE42831, GSE42832, GSE83456 and GSEBruno [17]. To enhance the comprehensiveness of this study, a focused platform was considered. Specifically, only Platform-GPL10558 Illumina Microarray was exclusively utilized to mitigate batch effects, and peripheral whole blood samples were selectively chosen as the primary biological material utilized for exploring the differential gene expression profile between different conditions (Appendix A).

### 2.3. Data Processing and Differential Gene Expression Analysis

The identification of differentially expressed genes (DEGs) across diverse TB conditions was conducted using the R package DESeq2 v1.38.2. This package facilitated a robust differential gene expression analysis on bulk RNA-seq data obtained from different conditions. To prepare the data for downstream analysis, RNA-seq counts were normalized, and variance stabilizing transformation (VST) was applied. The default Wald test in DESeq2 was employed for differential expression analysis, and *p*-values were adjusted using the Benjamini–Hochberg method. Genes matching the criteria of an adjusted *p*-value less than 0.05 and an absolute fold change greater than 1 were considered DEGs. Logarithmically transformed data were computed with DESeq2, and batch effects were eliminated using the R package limma v3.44.3. Following batch effect correction, the data underwent principal component analysis (PCA) and weighted correlation network analysis (WGCNA). This comprehensive approach ensured the precise selection of genes displaying significant expression alterations across various TB conditions, forming a foundation for subsequent analyses. To visualize the results, a volcano plot illustrating the relationship between fold change and statistical significance was generated. Additionally, a heatmap, created using the ComplexHeatmap R package, provided a global view of gene expression levels across conditions. The top upregulated genes were clustered based on Euclidean distance, and each gene cluster underwent enrichment analysis using MSigDB 2023 gene sets. The results were presented through a scatterplot depicting odds ratio (x-position) and −log10 (*p*-value) (y-position). The entire analysis, encompassing the construction and interpretation of visualizations such as volcano plots and heatmaps, was executed using the latest version of R.

### 2.4. Reference-Based Decomposition

The R toolkit, Bisque, was employed for reference-based decomposition to accurately and efficiently estimate cell composition from bulk expression data using a single-cell reference. This method leverages single-cell data for the decomposition of bulk expression, implementing a regression-based approach that utilizes scRNA-seq or single-nucleus RNA-seq (snRNA-seq) data. Bisque generates a reference expression profile and learns gene-specific bulk expression transformations, enabling robust decomposition of RNA-seq data. To enhance the precision of the analysis, we utilized a newly integrated single-cell reference of human peripheral blood specifically developed for this study.

### 2.5. Peripheral Immune Cell and NK Cell Reference Map

scRNA-seq data from 30 different studies and 100 samples for total of 160K high-quality cells were integrated (Appendix A). scRNA-seq analyses were performed using Seurat (Version 4.3.0) and SingleR (Version 2.0.0) [18]. Quality control was primarily evaluated based on the number of feature genes and the expression percentage of mitochondrial genes. To accurately identify the different immune cell subsets. Cells that had more than 1000 detected genes and had less than 10% mitochondrial gene expression were considered high-quality. Cells with more than 10% mitochondrial gene expression were excluded. Gene counts were normalized with the NormalizeData function of Seurat and all the cells were integrated using RPCAIntegration. Intergraded data from all samples were clustered with 50 PCs in combination with the dimensional reduction method of uniform manifold approximation and projection (UMAP). Cell type annotation was performed with ScType [19], and those cells annotated to be NK cells were extracted for subsequent analyses. The Seurat package was used to calculate the feature genes of NK cell subsets, while the single-cell atlas of peripheral NK cells was utilized as a reference map to estimate cell composition (“decomposition”) from bulk expression data with single-cell information.

### 2.6. Preparation of PBMCs

PBMCs were isolated from blood samples using a conventional Ficoll-Paque density gradient centrifugation protocol in tubes containing EDTA. Then, PBMCs were washed and resuspended in RPMI 1640 medium supplemented with 10% FBS, penicillin (100 U/mL)–streptomycin (100 μg/mL) (all purchased from Sigma-Aldrich, Saint Louis, MO, USA). PBMCs were counted in Trypan blue, collected into flow cytometry tubes, and then washed with 1 mL of BD staining buffer.

### 2.7. Staining of Surface Antigens for Flow Cytometry

PBMCs (10^6^ cells) were aliquoted into flow cytometry tubes and monoclonal antibodies (mAbs) to CD3 (PerCP-Vio700-conjugated, clone REA613, Miltenyi Biotec, Koto City, Japan), CD19 (VioBlue-conjugated, clone REA613, Miltenyi Biotec), CD16 (PE-Cyanine7-conjugated, clone REA613, Miltenyi Biotec), CD56 (PE-conjugated, BD Bioscience San Jose, CA, USA), CD14 (PerCP-conjugated, BD Bioscience San Jose, CA, USA), NKG2C (FITC-conjugated, clone REA613, Miltenyi Biotec), CD127 (APC-Vio770-conjugated, clone REA613, Miltenyi Biotec), CD52 (FITC-conjugated, clone REA164, Miltenyi Biotec) were added for cell surface antigen staining. After incubating for 30 min in the dark at room temperature, the cells were washed twice with 1 mL of BD Staining Buffer (PBS without Ca^2+^ and Mg^2+^, 1% FBS, 0.09% sodium azide) and resuspended in 300 µL of BD Staining Buffer before being analyzed using flow cytometry. Samples were run on a BD FACS Lyric^TM^ flow cytometer, and data were evaluated with BD FACSuite™ V1.5 Application (BD Biosciences, San Jose, CA, USA) after collecting 200,000 gated events (lymphocytes). Peripheral blood lymphocytes were gated using forward (FSC) and side scatter (SSC) parameters, single cells and live cells. NK cells were identified in the CD3-negative, CD19-negative, CD14-negative and CD127-negative cells to exclude T cells, B cells, monocytes and ILCs, respectively, referred to as lineage (Lin)-negative cells, expressing CD16 and CD56 surface markers. Relevant isotype controls were also used.

### 2.8. Statistical Analysis

The statistical analysis was performed using R software version 4.0.3. The Wilcoxon test was used to compare continuous variables between two groups, whereas the Kruskal–Wallis test was used to compare continuous variables among three or more groups. A test was considered statistically significant if the *p*-value was less than 0.05. For data analysis and visualization, we used the R packages ggplot2, ggstatsplot and ggpubr [20].

## 3. Results

### 3.1. DEG and GSEA Analysis Reveal Specific Enrichment of NK-Mediated Immune Responses in LTBI Subjects

Since peripheral blood from individuals with different Mtb infectious statuses could exhibit distinct transcription profiles, we used RNAseq data from 1467 samples (TB = 896, LTBI = 298, HD = 273) to analyze gene expression and functional enrichment in the peripheral blood of TB patients and LTBI subjects. In addition, we also included RNAseq data from 633 OD samples. To this aim, we utilized differential gene expression analysis alongside the Gene Ontology Biological Processing (GO) gene set (Figure 1A). This approach allowed us to assess a diverse array of biological responses, particularly focusing on immune responses. By employing these methodologies, we gained insights into the intricate molecular landscapes associated with TB and LTBI conditions, shedding light on the specific genes and biological processes that contribute to the observed differences in the peripheral blood profiles. We determined the fold changes of gene expression levels, and the ratios of the fold changes were compared among three pair-wise comparisons (LTBI versus HD; TB versus LTBI; and TB versus HD) (Figure 1B). Transcriptional profiles of LTBI and HD samples exhibited relatively similar patterns, with 242 differentially expressed genes being observed, while TB samples exhibited much more differentially expressed genes when compared with LTBI and HD samples (*n* = 1786 and 1866, respectively; Appendix A). Differentially expressed genes in these three pair-wise comparisons with ratio >3 are presented in Appendix A. To identify shared transcriptional patterns among HD, LTBI and TB samples, we employed a Venn diagram analysis. This method allowed us to visualize the overlap in gene expression changes between HD versus TB and LTBI versus TB. Notably, our findings revealed that 611 genes exhibited a consistent differential expression across these pair-wise comparisons. Interestingly, this shared pattern was predominantly associated with genes related to NK cell activity. The significant representation of NK-related genes suggests a potential role of NK cells in the immune response against Mtb.

To elucidate the specific pathways within the shared genes, we conducted an enrichment analysis utilizing 611 identified genes and referencing the human KEGG pathway database. The analysis revealed significant enrichment in the NK-cell-mediated cytotoxicity pathway among these shared genes (*p*-value < 0.001 and odds ratio > 1000) (Figure 1C). This finding underscores the pronounced involvement of NK cell activities within the identified gene set, shedding light on their potential role in the immune response associated with these genes.

Lastly, we curated a signature of NK-cell-related genes by integrating information from nine distinct reference NK signatures (Appendix A). Of the 611 shared genes, only 26 were identified to be part of our NK cell signature dataset. Employing these genes, we constructed a heatmap, providing a visual representation of their expression patterns across all samples. Intriguingly, our findings unveiled a discernible separation between individuals with LTBI and HD based on these NK-cell-related genes (Figure 1D). Notably, key genes driving this separation included *SH2D1B*, *KLRF1*, *PRF1* and *GZMB*, indicating their significant involvement in the distinctive molecular profiles observed in LTBI and HD subjects. In the TB immune landscape and NK cell interactions, *SH2D1B* (SH2 Domain Containing 1B) and *KLRF1* emerge as pivotal players. *SH2D1B*, also known as *EAT-2*, is implicated in the immune response against Mtb, suggesting its potential role in shaping the host defense mechanisms during TB infection. On the other hand, *KLRF1*, a gene associated with NK cells, contributes to the expansion of specific NK cell subsets, such as NKp46^+^CD27^+^KLRG1^+^ cells, observed in LTBI individuals in an IL-21-dependent manner [6]. These insights into the functions of *SH2D1B* and *KLRF1* provide valuable clues to the nuanced dynamics of NK cell involvement in TB.

### 3.2. Computational Exploration into Immune Cell Composition in Different TB Conditions

Understanding the immune cell composition in individuals with TB is pivotal for developing effective biomarkers that can help in monitoring TB treatment progress and informing clinical decisions [21]. A comparative analysis of gene expression profiles among diverse TB subjects holds the key to unraveling phenotypes unique to LTBI. Notably, variations in the composition of immune cells in peripheral blood emerge as a pivotal factor in evaluating the activity of the immune system.

Examining immune cell populations at the single-cell level provides invaluable insights into their dynamic changes, especially in the context of TB. However, it is noteworthy that a comprehensive single-cell sequencing immune landscape specific to TB is currently unavailable.

In pursuit of a comprehensive understanding of immune cell composition in TB, we embarked on a computational exploration, recognizing the pivotal role of deciphering distinct phenotypes, particularly in LTBI. To delve into the intricate landscape of TB-related immune cell variations, we curated a comprehensive scRNA-seq reference panel from 110 different studies, encompassing a diverse array of subjects containing a total of 160K cells scRNA-seq data from 30 different studies, including a total of 100 subjects. Leveraging this extensive integrated dataset, PBMC cells were aligned and projected in two-dimensional space through uniform manifold approximation and projection (UMAP) to allow the identification of cell populations. Unsupervised clustering and canonical marker gene assessment generated eight major cell clusters (T lymphocytes, B lymphocytes, NK lymphocytes, monocytes, neutrophils, MAIT cells and γδ T cells) (Figure 1A). This reference panel served as a valuable resource for decomposing bulk expression data from 15 distinct datasets, encompassing a total of 2100 samples across 4 clinical groups. These groups included HD (*n* = 273), LTBI (*n* = 298), TB (*n* = 896) and OD (*n* = 633). Notably, 645 samples from HIV-positive individuals were excluded from the analysis, resulting in a final dataset of 1467 samples. To ensure data integrity, batch effect correction was employed to mitigate variations introduced by different studies. This method allowed us to dissect the complex gene expression profiles associated with different TB phenotypes. Through the decomposition of bulk expression data, our analysis revealed noteworthy findings. Specifically, among the examined subsets, only three demonstrated significant changes. Most notably, NK cells exhibited a substantial increase in the LTBI samples, with a statistically significant *p*-value < 0.01 (Figure 2B). In contrast, other cell subsets, such as monocytes and γδ T cells, displayed minimal alterations across the conditions. The remaining six clusters exhibited only minor and no statistically significant differences between all tested groups (Figure 1). The marked increase in NK cell abundance in LTBI, compared to HD and TB, suggests the potential of NK cells as a distinguishing biomarker for LTBI.

### 3.3. Comparison of NK Cell Transcriptional Landscape in Peripheral Blood of Different TB Conditions

Leveraging previous results, we virtually sorted all NK cells from the obtained maps, generating an additional NK reference map. This specialized map allows us to focus specifically on the dynamic changes within NK cell subsets across the spectrum of TB conditions. By re-clustering around 16,000 NK cells, we aim to unravel the nuanced variations in their phenotypic and functional profiles during different stages of Mtb infection. After new dimensional reduction on virtually sorted NK cells, we found six distinct NK subclusters (Figure 3A), based on the differential expression of canonical genes (Figure 3B). The largest cluster (C4) consisted of cytotoxic NK cells (*FCGR3A*, *FCER1A* and *SPON2*), and a second smaller cluster (C3) consisted of similar cytotoxic NK cells (*FCGR3A* and *SPON2*), which also expressed genes related to an antiviral state (*IFI6*, *IFI27*, *IFITM3* and *MX1*). The second most represented cluster (C5) consisted of CD52-positive NK cells (*KLRC2*, *CD52*, *CCL5* and *HLA-DRB1*), which most likely correspond to memory-like NK2.1 cells, which accumulate with age, exhibit proinflammatory characteristics and display a type-I interferon response state [22]. Cluster C2 included cytokine NK cells (*CCL4* and *IFNG*) [23], while cluster C6 expressed genes related to immune response regulation (*CD38*, *IFI44*, *IFI44L* and *GZMB*). Finally, the smaller cluster C1 included CD56^bright^ NK cells (*SELL*, *IL7R* and *GZMK*) [24].

Next, we analyzed the distribution of the six NK cell subtypes across different groups. Cluster C2 was significantly reduced in active TB patients, as compared to HD and LTBI subjects and OD patients. Conversely, cluster C5 showed an opposite behavior, as it was significantly increased in LTBI individuals, as compared to other groups (Figure 3C). Clusters 1, 3, 4 and 6 showed similar distribution among all tested groups.

To validate our in silico results, we applied flow cytometry analysis to study peripheral blood NK cells across the different groups. Figure 4A shows the FACS gating strategy of one sample per group, while Figure 4B shows the cumulative data from TB patients, LTBI subjects and HD. As shown in Figure 4B, NK cells, identified as described under Materials and Methods, and herein reported as CD16^+^ CD56^+^ cells for the sake of simplicity, among PBMC, were increased in LTBI subjects, as compared to TB patients and HD individuals, although significance was achieved only with this latter group. CD52^+^ NK cells (Figure 4C) were increased in LTBI subjects, but the only difference with active TB subjects was statistically significant. Most interestingly, the NKG2C^+^ (Figure 4D) and NKG2C^+^ CD52^+^ (Figure 4E) subsets were dramatically increased in LTBI individuals, as compared to both HD subjects and active TB patients, and differences attained statistical significance in all cases. Altogether, the flow cytometry data, although performed on a small number of samples, fully reflect our previous in silico results related to the NKG2C^+^ CD52^+^ NK subset.

## 4. Discussion

Understanding the nature of protective immune responses to Mtb, as well as identifying biomarkers that may distinguish LTBI subjects from patients with active TB disease and these latter from patients with pulmonary diseases other than TB, is essential for the management of TB worldwide. While the importance of conventional CD4 and CD8 T cells in immune responses to Mtb has been well established, there is recent evidence of the participation of other unconventional lymphoid cells. In this regard, several studies have revealed the importance of NK cells, although their contribution to the control of Mtb infection is still unclear. Similarly, studies on the frequency and phenotype of NK cells in the peripheral blood of patients with active TB disease or LTBI subjects have yielded sometimes contrasting results.

In the present study, we have combined scRNA-seq and bulk RNA-seq data of hepatocellular cancer to analyze the molecular characteristics of circulating NK cells at different stages of Mtb infection, i.e., active TB disease versus LTBI, and compared these characteristics with those of HD and OD patients. We found that an NK cell cluster was significantly different among all four tested groups. In line with previously reported results from both phenotypic and molecular studies, we found that NK cells were decreased in active TB patients, probably reflecting the lymphopenia characteristic of TB [25]. Thus, our results are in agreement with another study that showed that NK-associated genes were significantly downregulated in active TB patients compared with LTBI subjects [26]. These findings suggest that NK cells contribute to the control of Mtb infection and that there are significant quantitative modifications in the NK cell population according to different infection conditions. 

However, and most notably, the NK cell cluster was significantly increased in LTBI subjects, as compared to patients with active TB or OD and HD. 

Since alterations in the frequency of circulating lymphocytes have not previously been uniquely associated with LTBI status, we became interested in further investigating the composition of different NK cell subclusters in the four tested groups. We re-clustered 16,000 peripheral blood NK cell transcriptomes extracted from the 160,000 PBMC scRNA-seq data from 30 different studies including a total of 100 subjects. UMAP analysis identified six distinct subclusters (Figure 3A), based on differential expression of canonical genes (Figure 3B), corresponding to well-known NK cell subtypes. An analysis of NK subcluster distribution across different tested groups revealed that only two clusters, C2 and C5, significantly changed between LTBI and all other groups. Cluster C2 was significantly reduced in LTBI subjects, as compared to HD, TB and OD patients. Conversely, cluster C5 showed an opposite behavior, as it was significantly increased in LTBI individuals, as compared to other groups. Clusters 1, 3, 4 and 6 showed similar distribution among all tested groups. Cluster C2 included cytokine NK cells (CCL4 and IFN-γ) [23], while cluster C5 consisted of CD52^bright^ NK cells (KLRC2, CD52, CCL5 and HLA-DRB1), which most likely correspond to memory-like NK2.1 cells, which accumulate with age, exhibit proinflammatory characteristic and display a type-I interferon response state [23]. NK2.1 cells have been shown to reduce the expression levels of the FCGR3A and FCER1A genes and elevate the expression of *KLRC2* (NKG2C), among others. Since it has been reported that *KLRC2* (NKG2C) is the hallmark of NK2.1 cells and can be used to distinguish NK2.1 subsets from other NK subsets, we have used flow cytometry to confirm the expansion of this subset in LTBI subjects. As shown in Figure 4, we confirmed in a small group of peripheral blood samples that LTBI subjects had significantly higher percentages of CD56^dim^ CD16^bright^ NKG2C^+^ NK cells, than active TB patients. However, differences between LTBI subjects with HD did not attain statistical significance, probably because of the small cohort samples.

In humans, infections with Cytomegalovirus (CMV) [27], hepatitis B and C virus [28], hantavirus [29] and Chikungunya virus (CHIKV) [30] lead to imprinted NK cell receptor repertoires with increased frequencies of specific NK cell subsets. Interestingly, in CMV infection, NKG2C^+^ NK cells are elevated during the acute phase of the disease, and the level is then sustained for a year post-infection [31]. In response to CHIKV infection, the repertoire of activating and inhibitory NK cell receptors is modulated and the increase in NKG2C^+^ NK cells correlates with viral load [30,32].

“Adaptive-memory” NKG2C^+^ NK cells are increased during CMV infection [32,33]. Similarly, and relevant to our study, NKG2C expression was increased in tuberculin skin test (TST)-positive (most likely latently infected) individuals, as compared to active TB patients and HD [26,34]. In our study, we did not find statistically significant differences in IgG antibody titers to several common pathogenic viruses including EBV, HSV-1/2, VZV and CMV, excluding the possibility that differences reported may be attributable to preexisting viral infections. NKG2C expression highlights the adaptive nature of NK cells during chronic diseases, i.e., the specific expansion of an NK cell subset upon antigen re-exposure. NKG2C is an activating receptor that binds to HLA-E [35] and is expressed mainly on CD56^dim^ NK cells [36]. NK cells expressing NKG2C expand during viral infection [27,28,32] and have enhanced effector functions [37]. Our molecular and phenotypic results are in full agreement with previous studies [26,37], showing that the proportion of NKG2C^+^ CD56^dim^ CD16^−^ NK cells is significantly elevated in tuberculin skin test-positive healthy individuals (presumably LTBI subjects). The increased frequency of imprinted NK cell memory in LTBI individuals could be the result of continuous exposure to Mtb antigens, suggesting that this cell subset might be somehow involved in the control of Mtb infection at a latency stage and Mtb reactivation. Accordingly, higher adaptive NK cell expansion is associated with better disease-free survival after bone marrow transplantation [38].

While this work has limits due to the relatively low number tested phenotypically (i.e., by flow cytometry analysis), we believe it has merits since, to our knowledge, this is the first study combining scRNA-seq and bulk transcriptomics on very large datasets to identify changes in immune cell composition in human TB. Overall, such an approach that has been found recently helpful in cancer [39,40] may provide new opportunities for the evaluation of biomarkers and/or correlates of protection in human TB.

## Figures and Tables

**Figure 1 cells-13-00293-f001:**
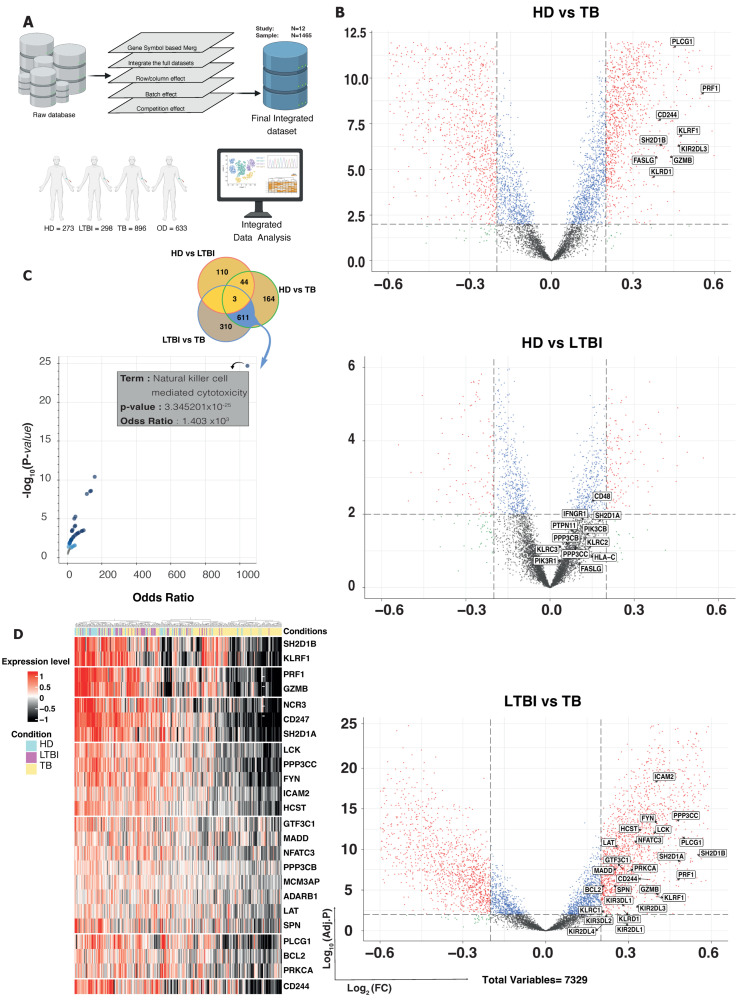
Integration of RNAseq data and gene expression analysis shows NK activity in TB. (**A**) Comprehensive RNAseq data, including HD (*n* = 273), LTBI (*n* = 298), TB (*n* = 896) and OD (*n* = 633). (**B**) Volcano plots illustrate gene expression variations in three key comparisons: HD versus LTBI, HD versus TB and LTBI versus TB. Each point on the plot represents a gene, with *x*-axis indicating log2 fold change and *y*-axis showing −log10 *p*-value. Only genes with significant regulation (≥0.2-fold change, *p*-value ≤ 0.001) are highlighted, focusing on upregulated genes related to the NK signature. A Venn diagram explores upregulated gene overlap in LTBI versus TB and HD versus TB. (**C**) Enrichment analysis on human KEGG pathways reveals shared significant pathways, depicted in a Volcano plot. Each point represents a term based on odds ratio (*x*-axis) and −log10 (*p*-value) (*y*-axis) from the overlapping gene set. Notably, NK-cell-mediated cytotoxicity is the most significantly enriched pathway shared between LTBI versus TB and HD versus TB. (**D**) A heatmap showcases gene expression profiles of significantly upregulated genes overlapping in LTBI versus TB and HD versus TB, specifically associated with NK-cell-mediated cytotoxicity.

**Figure 2 cells-13-00293-f002:**
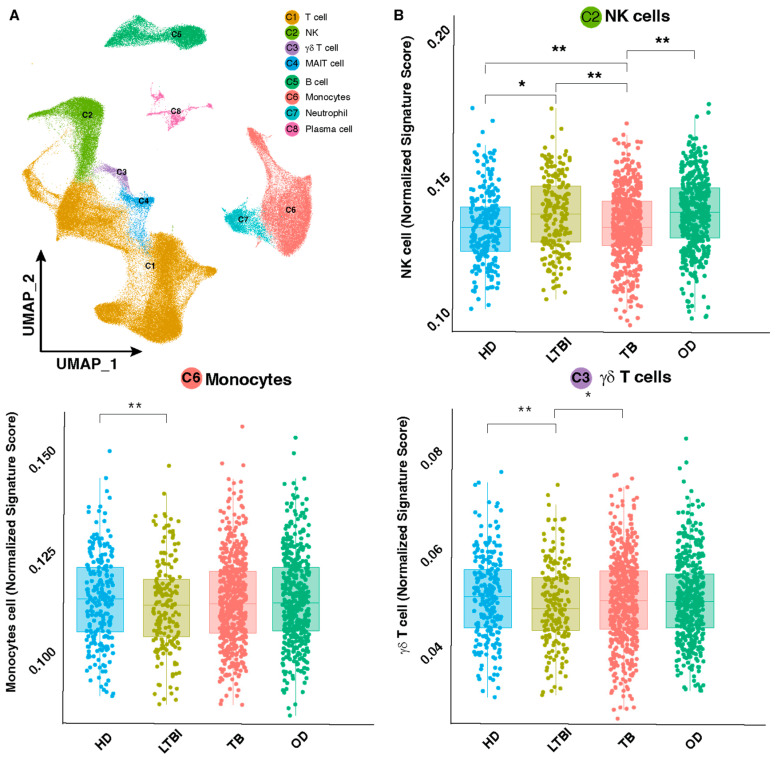
Virtual single-cell RNAseq shows a significant change in NK cells in different TB conditions. (**A**) A total of 160K cells from 30 different studies including a total of 100 subjects were integrated and, after normalization and dimensional reduction, led to identification of 8 different clusters. (**B**) The integrated PBMC map served as a guide for bulk expression deconvolution and revealed 3 cell subtypes significantly changed. Statistical significance was assessed using a two-way *t*-test, and *p*-values are denoted by symbols (** ≤ 0.01, * ≤ 0.05).

**Figure 3 cells-13-00293-f003:**
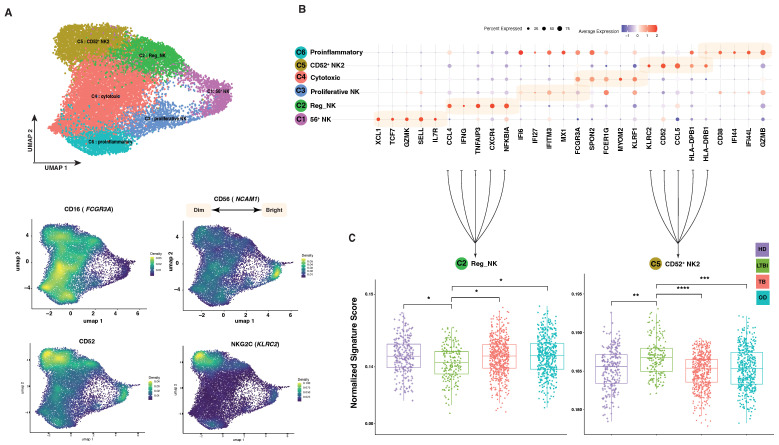
Dynamic changes in NK cell subsets during LTBI. (**A**) Dimensional reduction of virtually sorted NK cells demonstrated 6 different clusters based on the NK cell markers CD16 (*FCGR3A*), CD56 (*NCAM1*), CD52 (*CD52*) and NKG2C (*KLRC2*). (**B**) Dotplot of the top markers expressed in each NK cell type. (**C**) Bulk expression data decomposition using the new NK references revealed statistically significant alterations in NK subsets. The *p*-values, calculated through a two-way t-test, are represented by symbols (**** ≤ 0.0001 *** ≤ 0.001, ** ≤ 0.01, * ≤ 0.05).

**Figure 4 cells-13-00293-f004:**
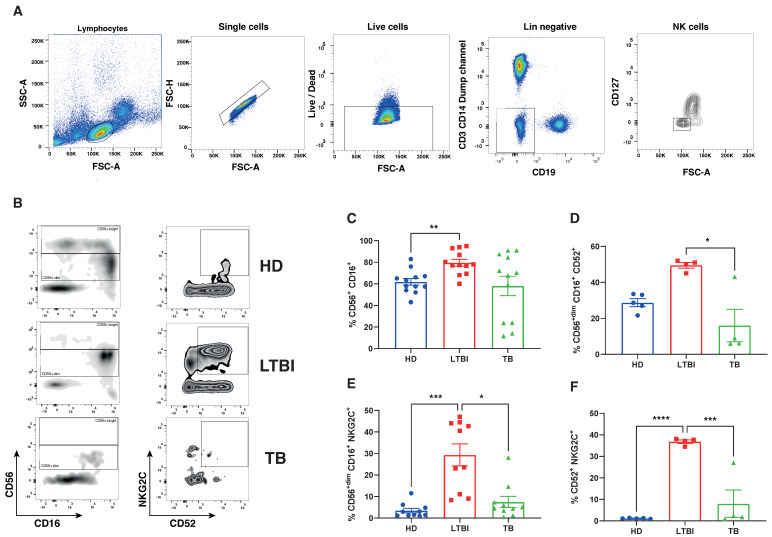
Flow cytometry analysis of NK cell subsets in peripheral blood of active TB patients, LTBI subjects and HD. (**A**) Gating strategy to access NK cell subsets: lymphocytes were gated using forward (FSC) and side scatter (SSC) parameters, single cells and live cells. NK cells were identified as CD3-negative, CD19-negative, CD14-negative, CD127-negative (Lin-negative) (**B**): cells expressing CD16 and CD56 along with the relative gating strategy used to identify memory-like NK2.1 cells. (**C**–**F**) Flow cytometry analysis of total NK (**C**), or NK cells expressing CD52 (**D**), NKG2C (**E**) and both CD52 and NKG2C (**F**), in PBMC of HD, LTBI subjects and TB patients. Each symbol represents one sample; bars represent mean with SEM values. *p*-values were calculated using the Kruskal–Wallis test, including multiple test correction. * *p* ≤ 0.05; ** *p* ≤ 0.01; *** *p* ≤ 0.001; **** *p* ≤ 0.0001.

## Data Availability

The raw data generated from both bulk and single-cell RNA sequencing are publicly available and all script and row cytometry data presented in this study are available upon request from the corresponding author.

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
