# Peer review of "Integrated Analysis of Single-Cell and Bulk RNA Sequencing Data Reveals Memory-like NK Cell Subset Associated with Mycobacterium tuberculosis Latency"

_cells, 2024, doi:10.3390/cells13040293_

Round 1
Reviewer 1 Report
Comments and Suggestions for Authors
This is an interesting study and useful addition to the literature.
Abstract, line 23, delete where appropriate: ‘we have combined we have combined’
Abstract, line 2, change to: ‘Altogether, our results provide new..’
Introduction: the authors should mention how NK cell functions are regulated by activating receptors that recognise sets of ligands expressed by stressed cells and infected cells and inhibitory receptors specific for MHC class I molecules and how this system works in terms of the ‘missing-self’ model (PMID: 2201309). As well as the latest breakthroughs in terms of recognition of soluble, secreted molecules by activating receptors (PMID: 29275861 & PMID: 25745066).
Results.
Figures and associated text need to be enlarged to make clearer.
I think the flow cytometry data currently in the supplemental data should be included as new Fig. 4 in the main manuscript.
Can the authors quantify the % of NK cell populations by flow cytometry from these patients and present as a bar graph?
Comments on the Quality of English LanguageA thorough English check would be appropriate.
Author Response
Abstract, line 23, delete where appropriate: ‘we have combined we have combined’
We have corrected the text eliminating misprints and errors.
Abstract, line 2, change to: ‘Altogether, our results provide new..’
We have corrected the text eliminating misprints and errors.
Introduction: the authors should mention how NK cell functions are regulated by activating receptors that recognise sets of ligands expressed by stressed cells and infected cells and inhibitory receptors specific for MHC class I molecules and how this system works in terms of the ‘missing-self’ model (PMID: 2201309). As well as the latest breakthroughs in terms of recognition of soluble, secreted molecules by activating receptors (PMID: 29275861 & PMID: 25745066).
Accepting the reviewer’s recommendation, we have modified the introduction to describe these important aspects of NK cell biology.
Results.
Figures and associated text need to be enlarged to make clearer.
Accepting the reviewer’s recommendation, we have improved the quality of figures and associated text.
I think the flow cytometry data currently in the supplemental data should be included as new Fig. 4 in the main manuscript.
Accepting the reviewer’s recommendation, we have moved the supplementary figure to the main manuscript as Figure 4.
Can the authors quantify the % of NK cell populations by flow cytometry from these patients and present as a bar graph?
Accepting the reviewer’s recommendation, we have now provided a bar graph (Figure 4) showing % NK cells in different tested groups.
Reviewer 2 Report
Comments and Suggestions for Authors
Dear authors, I read your manuscript with great interest. The paper has been done at a high professional level. The extremely small number of patients included in the study is the only drawback of the work (which does not diminish its value).
Author Response
Dear authors, I read your manuscript with great interest. The paper has been done at a high professional level. The extremely small number of patients included in the study is the only drawback of the work (which does not diminish its value).
We sincerely thank the reviewer for the very positive comment to our study. We agree with the reviewer that the limited number of patients under study is a limit to the overall significance of our results and we have briefly outlined this aspect.
Reviewer 3 Report
Comments and Suggestions for Authors
In this work, Mojtaba Shekarkar Azgomi et al. performed analysis of scRNA-seq and bulk RNA-seq data of peripheral blood NK cells from individuals with active pulmanory tuberculosis (PTB) and with latent tuberculosis infection (LTBI). This analysis allowed them to show more similar transcriptional pattern in NK cells associated with increased activity of cytotoxicity-mediated pathways between LTBI and healthy controls (HC) compared to PTB. Besides, 18 PBMC samples (6 in each group) were collected, and NKG2C expression level was analyzed in CD56bright/dim CD16+/-NK cells gated in PBMC by flow cytometry. Analysis of scRNA-seq data revealed expanded memory-like NK cells with the CD52+ phenotype. The results of the paper are new and interesting, although some issue should be addressed to improve the quality of the manuscript.
1. What is the percentage of the HCMV-seropositive individuals in the cohorts? It should be important to notice, as it might influence on the forming NKG2C+ memory-like NK cells.
2. 6 samples per group for NKG2C+ NK cell assessment is not enough to make definite conclusion, so the statement concerning NKG2C expression should be softened.
3. In the Discussion section, possible mechanisms for NKG2C+ NK cells involvement in TB protection would be good to discuss.
4. It looks questionable that final NK cell signature includes genes incoding not only NK cell receptors, but the ligands for NK cell receptors, such as MICA, MICB, ULBPs. Are NK cells expected to preferentially express such ligands?
5. Fig 1. In the figure legend numbers of individuals in groups should be clarified instead of dots.
6. Line 206-207. It is unclear, should be clarified: 1465 samples - TB = 4, LTBI = 4, HD = 4.
7. The manuscript should be checked for misprints, such as “Alltogether”, “a memory-like … NK cells” (page 1).
Comments on the Quality of English Language
The quality of English is acceptable, although the whole text should be checked carefully for typos.
Author Response
In this work, Mojtaba Shekarkar Azgomi et al. performed analysis of scRNA-seq and bulk RNA-seq data of peripheral blood NK cells from individuals with active pulmanory tuberculosis (PTB) and with latent tuberculosis infection (LTBI). This analysis allowed them to show more similar transcriptional pattern in NK cells associated with increased activity of cytotoxicity-mediated pathways between LTBI and healthy controls (HC) compared to PTB. Besides, 18 PBMC samples (6 in each group) were collected, and NKG2C expression level was analyzed in CD56bright/dim CD16+/-NK cells gated in PBMC by flow cytometry. Analysis of scRNA-seq data revealed expanded memory-like NK cells with the CD52+ phenotype. The results of the paper are new and interesting, although some issue should be addressed to improve the quality of the manuscript.
- What is the percentage of the HCMV-seropositive individuals in the cohorts? It should be important to notice, as it might influence on the forming NKG2C+ memory-like NK cells.
The point of the editor is correct. However, as we have stated in the text, we did not find statistically significant differences in IgG antibody titers to several common pathogenic viruses including EBV, HSV-1/2, VZV and CMV, excluding the possibility that differences reported may be attributable to preexisting viral infections.
- 6 samples per group for NKG2C+ NK cell assessment is not enough to make definite conclusion, so the statement concerning NKG2C expression should be softened.
We have now increased the number of tested subjects to 12 in each group and have also included staining with CD52. Accepting the reviewer’s recommendation, we have also toned down the statement concerning NKG2C expression.
- In the Discussion section, possible mechanisms for NKG2C+ NK cells involvement in TB protection would be good to discuss.
Accepting the reviewer’s suggestion, we have discussed the possible involvement of NKG2C NK cells in immune responses to Mycobacterium tuberculosis.
- It looks questionable that final NK cell signature includes genes incoding not only NK cell receptors, but the ligands for NK cell receptors, such as MICA, MICB, ULBPs. Are NK cells expected to preferentially express such ligands?
We would like to thank the reviewer for this insightful comment. It is indeed an important consideration whether NK cells are expected to express ligands such as MICA, MICB, and ULBPs. Our approach to defining the NK cell signature was comprehensive, namely the NK cell signature in our study was developed by integrating several established gene sets, including those from KEGG, MSigDB, and references such as Rooney et al (Ref)., Bindea et al( Ref ), and Danaher et al (Ref1,Ref2). These sources provide a broad and recognized basis for identifying NK cell-related genes. Our in-house developed gene set includes genes related to NK cell activity. The inclusion of genes encoding ligands for NK cell receptors, such as MICA, MICB, and ULBPs, is based on a comprehensive view of NK cell biology. In fact, NK cells interactions do not just occur through receptors, but also via these ligands. This interaction is crucial for the regulation of NK cell activity, including self-recognition and the initiation of immune responses, therefore, providing a deeper understanding of the potential regulatory mechanisms and interactions in which NK cells can be involved. Thus, our goal was to create a robust and inclusive signature that captures the diverse functionalities and interactions of NK cells, rather than limiting the scope to traditional receptor-based activities.
- Fig 1. In the figure legend numbers of individuals in groups should be clarified instead of dots.
We have corrected the text: we sincerely apologize for the poor presentation of the figure.
- Line 206-207. It is unclear, should be clarified: 1465 samples - TB = 4, LTBI = 4, HD = 4.
We have corrected the text: we apologize for the poor presentation of the results.
- The manuscript should be checked for misprints, such as “Alltogether”, “a memory-like … NK cells” (page 1).
We have corrected the text eliminating misprints and errors.